# Chloride Homeostasis in Neuronal Disorders: Bridging Measurement to Therapy

**DOI:** 10.3390/life15091461

**Published:** 2025-09-17

**Authors:** Daniele Arosio, Carlo Musio

**Affiliations:** Istituto di Biofisica, Consiglio Nazionale delle Ricerche, 38123 Trento, Italy; carlo.musio@cnr.it

**Keywords:** chloride homeostasis, NKCC(1)/KCC(2), ClC, GABA signaling, neurological disorders, brain diseases, epilepsy, Alzheimer’s disease, Huntington’s disease, autism spectrum disorder, therapeutic targets, precision medicine

## Abstract

Neuronal chloride (Cl^−^) homeostasis is fundamental for brain function, with disruptions increasingly recognized as pathogenic across neurological disorders. This review synthesizes evidence from preclinical models and clinical studies, integrating electrophysiological measurements, molecular analyses, imaging with genetically encoded sensors like ClopHensor, and behavioral assays. Key findings demonstrate that Cl^−^ dysregulation follows distinct patterns: (1) in epilepsy, KCC2 downregulation converts GABAergic inhibition to excitation, promoting seizures; (2) in Alzheimer’s disease (AD) models, pre-symptomatic KCC2 loss in hippocampus is observed, with KCC2 restoration reversing aspects of cognitive decline; (3) in autism spectrum disorders (ASD), developmental delays in GABA polarity shifts feature due to altered NKCC1/KCC2 ratios; and (4) in Huntington’s disease (HD), striatal neuron-specific Cl^−^ imbalances are linked to motor dysfunction. Methodologically, advanced tools—including subcellular Cl^−^ imaging and high-throughput drug screening—have enabled precise dissection of these mechanisms. Therapeutic strategies targeting Cl^−^ transporters (NKCC1 inhibitors like bumetanide, KCC2 enhancers like CLP290) show preclinical promise but require improved central nervous system (CNS) delivery and selectivity. These findings establish Cl^−^ homeostasis as both a biomarker and therapeutic target, necessitating precision medicine approaches to address heterogeneity in neurological disorders.

## 1. Introduction

For decades, neuronal excitability has been primarily understood in terms of action potential generation mediated by cationic conductances and synaptic transmission of glutamate and γ-aminobutyric acid (GABA) [1,2,3]. However, a growing body of evidence highlights the critical, and historically underappreciated, role of intracellular Cl^−^ concentration ([Cl^−^]_i_) as a fundamental regulator of neuronal function and a key determinant of brain health [4]. The precise maintenance of [Cl^−^]_i_ is essential not only for establishing the inhibitory nature of GABAergic neurotransmission, but also for modulating neuronal excitability and plasticity throughout development and in the mature brain. This delicate balance is achieved through the coordinated action of Cl^−^ channels and transporters, most notably the cation-chloride cotransporter NKCC1 and the potassium-chloride (K^+^-Cl^−^) cotransporter KCC2, which regulate Cl^−^ movement across the neuronal membrane. The electrochemical Cl^−^ gradient, maintained by these opposing actions (NKCC1 importing Cl^−^ and KCC2 extruding it), governs GABA and glycine response polarity, neuronal excitability thresholds, and activity-dependent plasticity [5,6].

Disruptions in this tightly controlled Cl^−^ homeostasis have emerged as a common pathological thread linking diverse neurological and psychiatric disorders, ranging from epilepsy and AD to ASD and HD [5]. Growing evidence suggests that Cl^−^ dysregulation is not merely a consequence of these diseases, but rather a contributing factor to their initiation and progression, often manifesting early in the disease course. This realization is shifting the paradigm, positioning Cl^−^ homeostasis as not only a potential biomarker for early intervention but also a compelling therapeutic target. Indeed, a growing body of research demonstrates that restoring or maintaining proper Cl^−^ homeostasis can reverse or mitigate neurological deficits in preclinical models. For instance, in epilepsy, reduced KCC2 function and increased NKCC1 activity can convert inhibitory GABAergic signaling into excitation, promoting ictogenesis [7,8]; in AD’s models, preclinical studies reveal pre-symptomatic loss of KCC2 in the hippocampus and prefrontal cortex [9], with restoration of Cl^−^ homeostasis reversing cognitive deficits; in autism spectrum disorders altered expression ratio between NKCC1 and KCC2 delays the developmental GABA shift [10,11]; and in HD’s models, differential Cl^−^ dysregulation is observed in D1 and D2 medium spiny neurons, contributing to selective neuronal vulnerability [12,13].

This review aims to synthesize the current understanding of the molecular mechanisms governing Cl^−^ homeostasis, critically evaluate the growing evidence for its dysregulation in various disease states, and explore emerging therapeutic strategies. We will highlight key findings regarding Cl^−^ dysregulation across multiple disorders, detail recent methodological advances—including genetically encoded Cl^−^ indicators like ClopHensor and SuperClomeleon enabling dynamic [Cl^−^]_i_ tracking with subcellular resolution—and assess the translational potential of current and developing therapies. We will also identify crucial knowledge gaps that warrant future investigation, particularly in understanding cell-type-specific dysregulation patterns revealed by single-cell technologies. By highlighting the convergence of fundamental neuroscience and translational medicine, we aim to show that targeting Cl^−^ homeostasis is a plausible therapeutic avenue with the potential to benefit defined patient groups, contingent on improved CNS exposure, selectivity, and biomarker-guided stratification.

## 2. Mechanisms Maintaining Neuronal Cl^−^ Homeostasis

[Cl^−^]_i_ is a critical determinant of neuronal function, influencing both the electrical properties of the neuron and the nature of synaptic communication. Maintaining this concentration within a narrow physiological range requires a sophisticated interplay of membrane proteins and cellular processes [14]. This delicate balance is achieved through the coordinated action of Cl^−^ channels and transporters (see Appendix A
Table A1, and Figure 1), which regulate the movement of Cl^−^ across the neuronal membrane [15].

### 2.1. Role of Cl^−^ Channels: Passively Conducting the Flow

Cl^−^ channels are transmembrane proteins that form pores through the neuronal membrane, allowing for the rapid, passive movement of Cl^−^ down their electrochemical gradient [16]. This movement is driven by the difference in both the electrical potential and the concentration of Cl^−^ across the membrane [5]. Unlike transporters, which actively pump ions against their electrochemical gradient using energy, channels provide a mechanism for rapid ion flux, crucial for the fast changes in membrane potential that underlie neuronal signaling [17,18]. Several types of Cl^−^ channels contribute to neuronal function.

**Ligand-gated Cl^−^ channels:** These channels open in response to the binding of specific neurotransmitters. The most prominent examples are GABA_A_ receptors [19,20] and glycine receptors [21,22,23]. GABA_A_ receptors, activated by GABA, the primary inhibitory neurotransmitter in the mature brain, are permeable to both Cl^−^ and bicarbonate ions [5,6]. The direction of Cl^−^ flow through GABA_A_ receptors is determined by the electrochemical gradient, specifically the difference between the Cl^−^ reversal potential (E_Cl_) and the resting membrane potential (RMP). In mature neurons with a low [Cl^−^]_i_, E_Cl_ is typically more negative than the RMP, leading to an influx of Cl^−^ upon GABA_A_ receptor activation, causing hyperpolarization and inhibition of neuronal activity [24]. Glycine receptors, activated by glycine, primarily mediate inhibitory neurotransmission in the spinal cord and brainstem and are also permeable to Cl^−^, contributing to similar hyperpolarizing effects.

**Voltage-gated** Cl^−^
**channels (ClC family):** These channels open and close in response to changes in membrane potential, participating in diverse cellular processes such as neuronal excitability regulation and cell volume control [25,26]. Different isoforms within the ClC family exhibit distinct tissue distribution and functional properties, suggesting specialized roles in neuronal Cl^−^ homeostasis [27]. Mutations in ClC channels are linked to a variety of genetic disorders, often termed channelopathies, including myotonia congenita, osteopetrosis, and Dent’s disease [28,29]. Notably, ClC family members have been implicated in neurological disorders, highlighting their crucial role in maintaining neuronal function [25,30]. For example, ClC-1 contributes to stabilizing the membrane potential in muscle tissue and is also found in some neurons, potentially reducing their excitability [31,32].

ClC-2, meanwhile, is involved in cell volume regulation and influences neuronal excitability, also in pathophysiological conditions [33,34]. For instance, loss-of-function mutations in the CLCN2 gene, which encodes ClC-2 channels, cause leukoencephalopathy with ataxia (LKPAT; MIM #615651) [35]. Moreover, recessive mutations in MLC1 or GLIALCAM genes, where GlialCAM is a secondary subunit of ClC-2 channel, play a strong pathogenetic role in the megalencephalic leukoencephalopathy with subcortical cysts (MLC) [36]. In addition, ClC-2 channels have been identified in motoneuron-derived cells modeling the spinal and bulbar muscular atrophy (SBMA), where their current alterations are ameliorated by the neuropeptide PACAP [37].

More recently, mutations in ClCN6 have been identified as a new genetic cause of neuronal ceroid lipofuscinosis (NCL), a group of serious neurodegenerative disorders from lysosomial accumulation [38].

**Calcium-activated Cl^−^ channels (CaCCs):** These channels are activated by increased intracellular calcium concentration and contribute to a variety of neuronal functions, including synaptic plasticity, sensory transduction, and regulation of neuronal excitability [39]. For instance, TMEM16B (Ano2) can mediate Cl^−^ influx in hippocampal pyramidal cells and inferior olive neurons, leading to hyperpolarization [40], and TMEM16A (Ano1) plays precise roles in excitation and functioning of nociceptive sensory neurons [41,42].

**Other Cl^−^ channels:** Several other types of Cl^−^ channels with diverse regulatory mechanisms and functions exist in neurons. These include pH-sensitive Cl^−^ channels, whose activity is modulated by intracellular pH [43,44], and channels that are permeable to other anions, such as the glutamate-activated Cl^−^ channel (EAAT4), which can function as a sodium-dependent Cl^−^ channel, potentially limiting excessive Purkinje cell firing in the cerebellum [5,45].

### 2.2. Role of Cl^−^ Transporters: Actively Maintaining Ionic Balance

Cl^−^ transporters are membrane proteins that actively move Cl^−^ across the neuronal membrane, often against its electrochemical gradient. This active transport draws on ATP-dependent processes or the gradient of other ions. Several key transporters determine [Cl^−^]_i_ and thereby influence neuronal excitability.

**Cation-chloride cotransporters (SLC12 family):** These transporters move Cl^−^ together with sodium and/or potassium in an electroneutral manner [46,47,48]. The Na^+^-K^+^-2Cl^−^ cotransporter 1 (NKCC1) primarily imports Cl^−^ into the cell, especially in immature neurons, thereby elevating intracellular Cl^−^ and contributing to the depolarizing actions of GABA during early brain development [49,50]. NKCC1 expression generally decreases as development proceeds but remains high in certain adult cells, such as cerebellar granule cells [5,51,52].

In contrast, the K^+^-Cl^−^ cotransporter 2 (KCC2) is the principal Cl^−^ extruder in mature neurons [53,54]. KCC2 establishes and maintains the low [Cl^−^]_i_ required for hyperpolarizing GABAergic inhibition, and its developmental upregulation underlies the shift from depolarizing to hyperpolarizing GABA responses [55]. Other K^+^-Cl^−^ cotransporters, including KCC1 and KCC3, also contribute to Cl^−^ transport and cell volume regulation. However, their specific roles in different neuronal populations are still being investigated [56,57]. In some contexts, KCC3 can partially compensate for reduced KCC2 function, albeit with distinct kinetics and subcellular distribution.

NKCC1 and KCC2 are reciprocally controlled by the Cl^−^-sensitive WNK–SPAK/OSR1 kinase cascade and by activity- and injury-related pathways, which together tune transporter phosphorylation, trafficking, and surface stability; see Section 2.4 for details.

**Cl^−^/HCO_3_^−^ exchangers (SLC4 and SLC26 families):** These transporters exchange Cl^−^ for bicarbonate ions, linking intracellular pH regulation and Cl^−^ homeostasis [58,59,60]. Examples include SLC4A3 (AE3), which can act as a Cl^−^ accumulator, and SLC4A10 (NCBE), which mediates sodium-dependent Cl^−^ efflux while allowing bicarbonate influx and has been linked to neurological disorders [61].

Loss-of-function NCBE variants have been associated with neurodevelopmental disease with impaired GABAergic transmission [62], and disruption of SLC4A10 has been reported in frontal lobe epilepsy with cognitive impairments [63]. While interactions with SLC12 transporters are not fully characterized, AE/NCBE pathways couple pH and Cl^−^ regulation and may functionally interact with cotransporters under high network activity or metabolic stress. Consistent with this, AE3 and NKCC1 jointly contribute to Cl^−^ accumulation at GABAergic synapses in embryonic motoneurons [64].

The interplay between Cl^−^ and bicarbonate transport highlights the interconnectedness of distinct ionic regulation mechanisms within neurons.

### 2.3. Context Influencing Cl^−^ Homeostasis

[Cl^−^]_i_ is dynamically shaped by developmental stage, neuronal activity, neurotrophic factors, and the cellular milieu. During brain maturation, the relative expression and activity of NKCC1 and KCC2 undergo significant changes, leading to the developmental GABA shift [50,65]. High levels of neuronal activity can drive Cl^−^ influx through ligand-gated channels, transiently altering [Cl^−^]_i_ [66,67]. Neurotrophic factors, particularly the brain-derived neurotrophic factor (BDNF), modulate transporter expression and function. The effects of BDNF on KCC2 and NKCC1 expression and function vary depending on developmental stage and neuronal integrity. BDNF generally supports KCC2 upregulation during development, whereas in the mature or injured CNS, its effects are context-dependent and can be downregulatory following neuronal insult [68,69,70]. Beyond BDNF, other trophic pathways can modulate chloride transporters. Notably, vascular endothelial growth factor (VEGF) preserves inhibitory strength by preventing the axotomy-induced downregulation of KCC2 in adult extraocular motoneurons, whereas BDNF does not confer this protection in the same context [71]. This highlights that trophic control of NKCC1/KCC2 is pathway- and cell-type–specific. Oxidative stress and inflammation further modify transporter expression and function and can contribute to the disruption of Cl^−^ homeostasis in various brain disorders [72].

### 2.4. Integration and Interplay Between Channels and Transporters

Passive Cl^−^ fluxes through ligand-gated (GABA_A_/glycine) and voltage-gated channels interact continuously with active transport by SLC12 cotransporters to set [Cl^−^]_i_ in a state- and activity-dependent manner. During intense synaptic activity, transient Cl^−^ loading via GABA_A_/glycine receptors can outpace extrusion, shifting E_Cl_ until KCC2-driven clearance restores the gradient; conversely, robust KCC2 function stabilizes inhibitory efficacy during network bursts.

At the core of this coupling is the Cl^−^-sensitive WNK–SPAK/OSR1 kinase cascade, which reciprocally regulates SLC12 transporters: inhibitory phosphorylation restrains KCCs, while activating phosphorylation stimulates NKCCs [73]. WNK kinases themselves sense cytosolic Cl^−^, translating activity-dependent Cl^−^ transients into transporter phosphorylation and surface stability [47,48,67]. In brain tissue, WNK1 activates NKCCs via SPAK/OSR1-dependent phosphorylation while inhibiting KCCs, providing bidirectional control implicated in cerebral ischemia and spinal cord injury [74]. KCC2 regulation is further shaped by a protein–protein network (e.g., ATP1A2, CKB, Neto2, PKC), which modulates trafficking and transport activity [75].

Upstream modulators—including neuronal activity, BDNF/TrkB signaling, oxidative stress, and inflammation—alter KCC2/NKCC1 expression, phosphorylation status, and surface stability in a stage- and context-dependent manner [65,66,67,72]. These interactions explain how Cl^−^ homeostasis can compensate under modest perturbations yet tip into dysregulation when transporter reserve is depleted or channel drive is excessive.

## 3. The Dual Role of Cl^−^ in Neuronal Function and Signaling

Cl^−^ are fundamental to establishing and maintaining the neuron’s electrochemical gradient, a prerequisite for all forms of neuronal communication. [Cl^−^]_i_ critically regulates neuronal excitability by dynamically shifting between inhibitory and excitatory roles, governed by developmental stage and pathophysiological context. This section details the multifaceted roles of Cl^−^ in neuronal function, from synaptic transmission to intracellular signaling.

### 3.1. Inhibitory Neurotransmission: Hyperpolarization

In mature neurons, the KCC2 cotransporter maintains a low [Cl^−^]_i_, so activation of GABA_A_ and glycine receptors drives Cl^−^ influx toward E_Cl_, resulting in membrane hyperpolarization.

This suppresses excitability via: (1) direct inhibition by shifting the membrane potential away from firing threshold, and (2) shunting inhibition, where increasing the membrane conductance increases the efficacy of excitatory inputs without necessarily causing a significant change in the membrane potential [76]. Overall, Cl^−^-mediated inhibition helps suppress excessive excitation.

### 3.2. Excitatory Neurotransmission: Depolarization

In contrast, during early development and in certain pathological states, GABA_A_ receptor signaling can be depolarizing and even excitatory. Predominant NKCC1 activity in immature neurons maintains elevated [Cl^−^]_i_, making E_Cl_ more positive than the resting membrane potential. When E_Cl_ exceeds rest, opening GABA_A_ receptors produces net Cl^−^ efflux (with additional HCO_3_^−^ efflux), depolarizing the membrane and supporting activity-dependent plasticity and circuit refinement. In adults, a similar depolarizing GABA response can re-emerge when Cl^−^ homeostasis is perturbed—for example, in primary afferent neurons during neuropathic pain—where elevated [Cl^−^]_i_ shifts E_Cl_ toward or above rest, yielding depolarizing GABAergic signaling implicated in pain processing [4,72].

### 3.3. Firing Threshold Control

The precise level of [Cl^−^]_i_ plays a critical role in regulating neuronal excitability and the generation of action potentials. E_Cl_ is highly sensitive to changes in [Cl^−^]_i_ and determines the direction and magnitude of Cl^−^ flow upon channel opening. Even small changes in [Cl^−^]_i_ can significantly alter E_Cl_ and consequently the strength and polarity of GABAergic and glycinergic responses, thus affecting how readily a neuron will fire an action potential [77]. Notably, research has demonstrated that Cl^−^ overload can lower the action potential threshold, making neurons more excitable [78,79]. This direct impact on the firing threshold reveals a potent mechanism by which Cl^−^ dysregulation can contribute to conditions characterized by neuronal hyperexcitability, such as epilepsy.

### 3.4. Involvement in Synaptic Plasticity and Neural Circuit Development

Beyond its immediate effects on neuronal excitability, Cl^−^ plays a crucial role in shaping synaptic connections and the development of neural circuits, particularly during early postnatal life [50]. The developmental GABA shift, driven by changes in Cl^−^ homeostasis, is essential for the maturation and refinement of these circuits in response to experience and environmental stimuli [65]. Emerging evidence also suggests that Cl^−^ is involved in the mechanisms underlying long-term potentiation (LTP) and long-term depression (LTD), the cellular bases of learning and memory [80,81].

### 3.5. Cl^−^ as an Intracellular Signaling Ion

Intriguingly, Cl^−^ are not solely involved in establishing membrane potentials and mediating synaptic transmission; they can also function as intracellular signaling molecules that directly modulate the activity of ion channels and transporters [82]. Notably, Heubl et al. [83] provided the first direct evidence that GABA_A_-evoked Cl^−^ influx acts as a regulatory second messenger by rapidly reducing KCC2 membrane diffusion and increasing its retention at synapses, thereby tuning inhibitory efficacy. Mechanistically, this rapid tuning is mediated by Cl^−^-sensitive WNK kinases, linking GABA_A_-driven Cl^−^ transients to KCC2 phosphorylation and trafficking [79,84]. More broadly, changes in intracellular Cl^−^ regulate cellular excitability via effects on NKCC1/KCC2 transporter activity [79]. This perspective broadens therapeutic opportunities; for example, the WNK–SPAK/OSR1 cascade has been proposed as a pharmacological and genetic target in related neurological disorders [84].

In addition, intracellular Cl^−^ can directly bind to and allosterically regulate the gating of certain voltage-gated Cl^−^ channels, such as ClC-0 and ClC-2 [26,29]. Furthermore, Cl^−^ can act as an allosteric modulator on channels that do not conduct it, such as potassium channels like SLO2 and non-selective cation channels like TRPM7. Cl^−^ can also modulate the activity of certain transporters, such as the Na^+^/HCO_3_^−^ cotransporter NBCe1, impacting intracellular pH homeostasis. These findings reveal a more complex role for Cl^−^ within neurons, extending beyond its classical involvement in electrical signaling and suggesting its participation in broader cellular regulatory pathways.

## 4. Methods to Measure Cl^−^ Homeostasis in Neuronal Systems

Studying the intricate role of Cl^−^ homeostasis in neuronal function and its disruption in various disorders necessitates the use of sophisticated techniques to measure [Cl^−^]_i_ and the activity of Cl^−^ channels and transporters. Several methods are currently employed, each with distinct advantages and limitations.

### 4.1. Electrophysiological Techniques: Recording the Electrical Signals

Electrophysiology provides direct measurements of the electrical activity arising from ion flow across neuronal membranes. Different patch-clamp configurations are currently well established to study Cl^−^ currents and estimate [Cl^−^]_i_ [85,86].

**Perforated patch clamp:** This technique uses antibiotics like gramicidin or amphotericin B to create small pores in the neuronal membrane, allowing electrical access to the cell without disrupting the intracellular Cl^−^ concentration [87]. This method is often used to measure the reversal potential of GABA- or glycine-evoked currents, which can then be used to estimate [Cl^−^]_i_ [88].

**Whole-cell patch clamp:** While this method allows for direct control of the intracellular environment, it can also dialyze the cell, potentially altering [Cl^−^]_i_ over time. However, it is valuable for studying the overall Cl^−^ conductance and the effects of various manipulations on Cl^−^ currents [37,89,90,91].

**Voltage-ramp and Interpolation methods:** These voltage-clamp approaches utilize the current-voltage relationship of GABA- or glycine-evoked Cl^−^ currents to estimate the Cl^−^ equilibrium potential and thus [Cl^−^]_i_. Studies have shown that the voltage-ramp method is more accurate in detecting changes in [Cl^−^]_i_ during prolonged GABA application compared to the interpolation method [14,92].

**Dual cell-attached patch recordings:** This non-invasive technique can be used to measure the driving force for Cl^−^ across the neuronal membrane [91,93].

### 4.2. Fluorescence Imaging Techniques: Visualizing Cl^−^ Dynamics

Fluorescent indicators offer a way to visualize and quantify changes in [Cl^−^]_i_ with high spatial and temporal resolution [94].

**MQAE (6-methoxy-N-ethylquinolium iodide):** This cell-permeant dye is quenched by Cl^−^, meaning its fluorescence intensity is inversely proportional to [Cl^−^]_i_ [95]. MQAE can be used with both conventional and two-photon microscopy to measure [Cl^−^]_i_ in various neuronal compartments [96]. Fluorescence Lifetime Imaging (FLIM) of MQAE provides a quantitative measure of [Cl^−^]_i_ that is independent of dye concentration [97].

**Genetically Encoded Cl^−^ Indicators (GECIs):** These sensors represent a major advancement, as they are genetically expressed in neurons, allowing for targeted and long-term monitoring of [Cl^−^]_i_ with high specificity and minimal invasiveness [98]. The development of GECIs began over 25 years ago with the serendipitous discovery of Cl^−^ sensitivity in green fluorescent protein (GFP) variants [99]. Since then, researchers have actively expanded their potential through site-directed mutagenesis, combinatorial site-saturation mutagenesis, and chimeragenesis.

SuperClomeleon (SClm): One prominent GECI is SuperClomeleon, a ratiometric sensor based on Förster Resonance Energy Transfer (FRET) between cyan fluorescent protein (CFP) and yellow fluorescent protein (YFP), where the FRET efficiency changes with Cl^−^ binding [97]. SClm has been widely used to examine changes in [Cl^−^]_i_ in brain slices and cultured neurons, proving to be a powerful tool for measuring physiological changes in intracellular Cl^−^. However, a known limitation of SClm is its sensitivity to intracellular pH (pH_i_), which can complicate the interpretation of Cl^−^ measurements [100,101,102].

ClopHensor: To address the challenge of simultaneous pH and Cl^−^ changes, the original ClopHensor construct was engineered, a novel fusion protein capable of independently and simultaneously measuring Cl^−^ and pH [103]. This original sensor was based on the fusion of the pH and Cl^−^-sensitive GFP mutant E^2^GFP with the pH and Cl^−^-insensitive monomer DsRed. Further optimization led to ClopHensorN, a new genetically encoded ratiometric Cl^−^ and pH sensor specifically optimized for the nervous system [104]. ClopHensor allows for the dynamic, simultaneous quantification of intracellular Cl^−^ and H^+^ concentrations under various conditions, even when both ion concentrations are changing concomitantly [105]. It achieves this by using specific excitation wavelengths and a family of calibration curves to account for pH-dependent Cl^−^ affinity [106,107,108,109,110].

Recent Advancements: The field continues to evolve with the exploration of new fluorescent protein templates, such as mBeRFP derived from EqFP578 [111] and mNeonGreen, which have also been engineered into improved Cl^−^ sensors like the ChlorONs [112]. Advances in directed evolution are also being applied to engineer GFP-based indicators with enhanced function [113]. Furthermore, the integration of GECIs with advanced imaging techniques like two-photon microscopy enables in vivo imaging of [Cl^−^]_i_ in brain slices and awake animals [107,111,112,113], providing unprecedented spatial and temporal resolution [100]. While not directly Cl^−^ indicators, recent developments in genetically encoded voltage indicators (GEVIs) like Vega, which exhibit higher photon budget and improved photobleaching half-life compared to prior indicators, illustrate the ongoing progress in fluorescent protein engineering towards greater signal-to-noise and stability in extended recordings [114].

### 4.3. Other Assays

Besides direct measurement of Cl^−^, other assays can provide insights into the activity of Cl^−^-related transporters and channels.

**Thallium flux assays:** These assays utilize the permeability of certain potassium channels and some SLC12 family transporters to thallium ions. Changes in intracellular thallium (Tl^+^) concentration, measured using fluorescent indicators, can reflect the activity of these channels and transporters [115,116]. In practice, Tl^+^-flux readouts provide a high-throughput surrogate for transporter/channel activity (including NKCC1-linked K^+^ movement), but they lack ion selectivity and can conflate upstream channel modulation with transporter function; therefore, orthogonal assays are recommended for mechanistic dissection [117].

**Sodium-dependent SLC Transporter Assays:** These assays measure Na^+^ influx associated with the activity of sodium-coupled Cl^−^ cotransporters like NKCC1, providing an indirect readout of transporter function [117]. Because they report on Na^+^ rather than Cl^−^, they can be confounded by changes in Na^+^ channels/transporters, membrane potential, or cell health; results should be corroborated with direct Cl^−^ measurements (e.g., MQAE/FLIM, GECIs) and/or electrophysiology for mechanistic conclusions.

### 4.4. Comparison Across Methodologies and Calibration

Electrophysiology provides gold-standard functional readouts of Cl^−^ driving force and ratio between the GABA reversal potential (E_GABA_) and E_Cl_ with millisecond resolution. Perforated-patch preserves native [Cl^−^]_i_ and is preferred to estimate E_GABA_/E_Cl_; whole-cell enables controlled internal solutions at the cost of dialysis-induced [Cl^−^]_i_ drift; voltage-ramp methods detect dynamic shifts more reliably than interpolation; dual cell-attached is minimally invasive but lower throughput [87,88,91,92,93].

MQAE intensity imaging is simple and sensitive but requires careful calibration (e.g., Stern–Volmer plots) and is affected by dye loading and photobleaching; FLIM-MQAE mitigates concentration/depth artifacts at the cost of specialized instrumentation [97]. SuperClomeleon enables ratiometric readout and cell-type targeting, but its readout is pH-sensitive and demands pH correction for absolute [Cl^−^]_i_ [100,102]. ClopHensor and its derivatives simultaneously report pH and Cl^−^, using excitation-ratio families and in situ calibration to decouple coincident pH and Cl^−^ changes, enabling quantitative mapping in slices and in vivo [103,104,105,107,108,109,110]. Notably, LSSmClopHensor, allowed us to monitor Cl^−^ accumulation concomitant to pronounced acidification during epileptic like discharges in mouse models [106]. In general, advanced techniques like two-photon imaging of GECIs are particularly well-suited for studying Cl^−^ dynamics in specific cell types within animal models of diseases. Thallium- and sodium-flux assays afford higher-throughput measures of transporteror channel activity but are indirect and should be paired with a Cl^−^-specific readout and-or electrophysiology for mechanism. Practically, SuperClomeleon offers straightforward transgenic deployment for longitudinal studies, whereas ClopHensor excels when concomitant pH dynamics or tissue heterogeneity complicate interpretation; electrophysiology remains essential when precise driving-force measurements or fast dynamics are required.

The choice of method depends on the specific research question, the neuronal system being studied, and the desired spatial and temporal resolution. Combining different techniques can often provide a more comprehensive understanding of Cl^−^ homeostasis in neuronal systems.

## 5. Neuronal Disorders Linked to Disruptions in Cl^−^ Homeostasis

A growing body of research has firmly established a link between the dysregulation of neuronal Cl^−^ homeostasis and the pathogenesis of a wide array of neurological and neurodevelopmental disorders [12,25]. These disruptions can arise from genetic mutations affecting Cl^−^ channels and transporters, as well as from acquired factors such as neuronal hyperactivity [118,119], inflammation [120], and neurodegenerative processes [121]. The mechanisms by which Cl^−^ homeostasis is impaired vary depending on the specific neuronal disorder (see Appendix A
Table A2).

### 5.1. Epilepsy

Epilepsy, a neurological disorder characterized by recurrent seizures, has been extensively linked to disruptions in Cl^−^ homeostasis. Genetic mutations in genes encoding Cl^−^ channels (e.g., CLCN1, CLCN2, CLCN3, CLCN4, CLCN6) and transporters (e.g., KCC2) have been directly implicated in various forms of epilepsy, including severe early-onset forms like infantile migrating focal seizures. In particular, gain- and loss-of-function mutations in the CLCN2 gene, which encodes ClC-2 channels, may contribute to, but not trigger, the development of idiopathic epilepsy [122]. Impaired function or altered expression of the Cl^−^ cotransporters KCC2 and NKCC1 are frequently observed in epilepsy [8,123]. A reduction in KCC2 activity or expression, or an increase in NKCC1 activity, can lead to an accumulation of intracellular Cl^−^, resulting in a depolarizing shift in E_GABA_ and a switch from inhibitory to excitatory GABAergic signaling, thus increasing neuronal excitability and promoting seizure generation [7,72]. Furthermore, during periods of high neuronal activity, such as during a seizure, the influx of Cl^−^ through activated GABA_A_ receptors can overwhelm the extrusion capacity of KCC2, leading to an activity-dependent accumulation of intracellular Cl^−^, which can further exacerbate neuronal hyperexcitability and contribute to the propagation and maintenance of seizure activity [124].

The shift towards depolarizing GABAergic signaling and the lowering of the action potential threshold due to increased intracellular Cl^−^ concentration lead to neuronal hyperexcitability and an increased propensity for synchronized neuronal firing, resulting in seizures [78].

### 5.2. Alzheimer’s Disease

Emerging research has implicated disruptions in Cl^−^ homeostasis in the pathogenesis of AD, a neurodegenerative disorder characterized by progressive cognitive decline. Mouse models carrying AD-related mutations show pre-symptomatic loss of the neuronal K^+^–Cl^−^ cotransporter KCC2 in hippocampus and prefrontal cortex, weakeningGABAergic inhibition and disturbing Cl^−^ balance. The extent of KCC2 loss inversely correlates with age-dependent increases in amyloid-β 42 (Aβ42), a key protein implicated in AD pathology, and acute administration of Aβ42 exposure reduces membrane KCC2, consistent with impaired Cl^−^ extrusion and network hyperexcitability [9]. Conversely, other studies report no change in total KCC2 after Aβ42 treatment but observe increased NKCC1 expression after prolonged exposure (e.g., 30 days post-injection in CA1), a mechanism that would likewise raise [Cl^−^]_i_ and weaken inhibition [125]. Divergent results likely reflect differences in exposure paradigm (acute vs. chronic), readouts (surface vs. total protein), brain region, and model system. Causality has been probed pharmacologically: enhancing KCC2 function (e.g., CLP290) restored spatial memory and social behavior in AD models, whereas KCC2 inhibition (e.g., VU0463271) impaired performance, linking defective Cl^−^ extrusion to cognitive decline in these settings [9]. Complementing the Aβ-centric view, several studies indicate that the amyloid precursor protein (APP) acts as a physiological regulator of KCC2. In rodent hippocampus and cortex, genetic loss or knockdown of APP lowers KCC2 mRNA and protein, reduces surface-resident KCC2, shifts E_GABA_ to more depolarized values, and increases network excitability; conversely, restoring APP or applying soluble APP-α rescues KCC2 abundance and inhibitory tone [126]. Mechanistically, APP has been proposed to influence KCC2 through both transcriptional and post-translational routes—via APP intracellular domain (AICD)-dependent effects on SLC12A5 transcription and by stabilizing KCC2 at the plasma membrane and preserving phosphorylation states associated with transport activity [126,127]. In AD-relevant conditions, oligomeric Aβ and disease-associated shifts in APP processing appear to disrupt these homeostatic actions, resulting in reduced KCC2 function and weakened synaptic inhibition; boosting KCC2 expression or activity partially restores E_GABA_ and improves synaptic and behavioral endpoints in preclinical models [127].

Overall, early AD hyperexcitability appears most consistent with impaired Cl^−^ extrusion via KCC2 loss, although NKCC1 upregulation may contribute in specific models and stages. Clarifying the relative contributions and regulation of KCC2 and NKCC1 by Aβ and other injury signals remains essential for developing mechanistically grounded neurotherapeutic strategies [128].

### 5.3. Autism Spectrum Disorder

ASD, a neurodevelopmental condition characterized by deficits in social communication and interaction, as well as restricted and repetitive patterns of behavior, has also been linked to disruptions in Cl^−^ homeostasis. The precise mechanisms of Cl^−^ dysregulation are still being investigated, but evidence suggests potential alterations in the expression or function of both NKCC1 and KCC2 in specific brain regions [128]. These imbalances could lead to a higher [Cl^−^]_i_ and a depolarizing shift in E_GABA_ during critical developmental periods, potentially affecting the normal maturation of neural circuits [10]. Studies in animal models of autism have shown abnormally high neuronal Cl^−^ levels from birth, potentially due to reduced activity of Cl^−^ transporters responsible for Cl^−^ extrusion [11,129]. Furthermore, the potential role of the birth hormone oxytocin in regulating neuronal Cl^−^ levels and influencing the expression of autism-like symptoms has been investigated [130,131].

The imbalance between excitatory and inhibitory signaling resulting from altered Cl^−^ homeostasis is believed to disrupt normal brain development and contribute to the core behavioral symptoms, including deficits in social communication and interaction, as well as restricted and repetitive patterns of behavior.

### 5.4. Huntington’s Disease

HD, a progressive neurodegenerative disorder characterized by motor, cognitive, and psychiatric disturbances, has also been associated with dysregulation of Cl^−^ homeostasis in the brain [12]. Studies in both animal models and human patients have revealed reduced KCC2 expression and function, as well as potentially increased NKCC1 activity, in the striatum and hippocampus. In mouse models of HD, alterations in the expression and function of NKCC1 and KCC2 have been observed in the hippocampus and striatum, leading to weakened inhibition and paradoxical excitatory actions of GABA [12]. Mutant Huntingtin protein may directly or indirectly interfere with the normal function of these Cl^−^ transporters, contributing to an increase in [Cl^−^]_i_ and altered GABAergic transmission. Notably, studies suggest a differential alteration of Cl^−^ homeostasis in D1 and D2 medium spiny neurons in HD, the primary neuronal population affected by the disease, potentially contributing to the selective vulnerability of D2 neurons in the early stages of the disease [12,13]. Restoring Cl^−^ homeostasis in specific neuronal populations, such as D2 neurons, has shown promise in rescuing motor deficits in HD mouse models, suggesting a potential therapeutic strategy [132].

The altered neuronal excitability in the striatum and other affected brain regions, resulting from Cl^−^ dysregulation, is thought to contribute to both the motor impairments, such as chorea, and the cognitive and psychiatric disturbances characteristic of the disease [12,13].

### 5.5. Other Neurological Disorders: A Common Thread of Dysregulation

Beyond these major disorders, disruptions in Cl^−^ homeostasis have been implicated in several other neurological conditions. In **neuropathic pain**, nerve injury can trigger a downregulation of KCC2 in dorsal horn neurons of the spinal cord. This reduction in the neuron’s ability to extrude Cl^−^ results in an elevated [Cl^−^]_i_, causing GABA to become depolarizing and contributing to the development of mechanical allodynia, a condition where normally non-painful touch is perceived as painful [41,42]. The increased excitability of pain-transmitting neurons in the spinal cord, driven by depolarizing GABA, leads to the amplification of pain signals and the perception of pain from normally innocuous stimuli, such as light touch (allodynia).

**Schizophrenia** has been associated with genetic variations in genes encoding Cl^−^ cotransporters and altered GABAergic signaling, suggesting a potential role for Cl^−^ dyshomeostasis in its pathophysiology [56]. Individuals with **Down Syndrome** exhibit evidence of altered GABAergic signaling and increased NKCC1 expression, indicating a potential role for Cl^−^ dysregulation in the associated neurological features [133,134].

**Hyperekplexia (startle disease)**, a rare genetic disorder, is caused by mutations in glycine receptor subunits [135], leading to impaired Cl^−^-dependent inhibitory glycinergic neurotransmission [5]. Certain forms of **ataxia** have been linked to mutations in genes encoding Cl^−^ channels and transporters, highlighting the importance of proper Cl^−^ regulation for cerebellar function and motor coordination [136].

Specific genetic mutations affecting Cl^−^ channels and transporters can result in severe forms of **neonatal seizures** and epilepsy, underscoring the critical role of Cl^−^ homeostasis in early brain development and function [137,138]. Impaired Cl^−^ homeostasis has also been associated with pathological processes following **acute brain injuries**, such as hypoxic–ischemic encephalopathy, brain edema, and post-traumatic seizures, contributing to neuronal swelling, excitatory GABA signaling, and increased seizure susceptibility [139,140].

**Rett Syndrome (RTT)**, a genetic disease caused by loss-of-function mutations in the MeCP2 gene, which alters the excitation-to-inhibition (E/I) ratio, leading to circuit-wide changes. Notably, an altered KCC2/NKCC1 ratio leads to a more depolarized E_GABA_ [141]. Finally, even subtle changes in Cl^−^ homeostasis can have significant consequences for neuronal coding and information processing, potentially contributing to the complex symptomatology of these disorders [142].

## 6. Emerging Research on Cl^−^ Dysregulation in Symptom Pathogenesis

Current research continues to investigate the intricate link between Cl^−^ dysregulation and symptoms across neuronal disorders. In AD, work focuses on early-stage KCC2 downregulation and its direct impact on cognition in animal models [9]. In HD, studies are dissecting striatal Cl^−^ dysregulation and testing whether restoring KCC2 function alleviates motor deficits [12]. In epilepsy, the role of Cl^−^ channels is receiving increasing attention, including contributions of ClC family members beyond cation channel mechanisms [7]. Investigations into the effects of Cl^−^ overload on neuronal excitability, particularly in hippocampal neurons, are clarifying mechanisms of hyperexcitability relevant to epilepsy [78]. The potential of oxytocin to modulate neuronal Cl^−^ levels and influence autism-related behaviors remains an active area of research [131]. Additionally, oxidative stress and inflammation are being explored as upstream regulators of Cl^−^ homeostasis across neurological conditions [72]. Together, these efforts underscore the dynamic nature of [Cl^−^]_i_ and its crucial role in neuronal network function in both health and disease [142].

Preliminary evidence suggests that sex and hormonal status may modulate NKCC1/KCC2 regulation and the timing of the GABA polarity shift. Although current data are limited and context-specific (e.g., strongest in pain circuits and select neurodevelopmental models), future studies should incorporate sex-stratified designs and quantitative Cl^−^ imaging to test for differences in transporter regulation and treatment response [143,144].

## 7. Potential Therapeutic Strategies Targeting Chloride Homeostasis

The growing recognition of Cl^−^ dysregulation in neurological disease has catalyzed therapeutic strategies aimed at restoring or modulating Cl^−^ homeostasis (see Appendix A
Table A3). Pharmacological modulation of SLC12 cotransporters is a major focus [145]. Inhibitors of NKCC1, such as bumetanide, are being investigated for their potential to reduce [Cl^−^]_i_ and restore inhibitory GABAergic signaling in disorders like epilepsy and ASD. Clinical translation remains challenging due to poor brain penetration and active efflux, motivating prodrugs, brain-targeted formulations, and alternative delivery routes; disease heterogeneity likely contributes to variable efficacy [9,146,147,148,149]. Among bumetanide derivatives, bumepamine exhibits substantially improved CNS exposure (approximately sevenfold higher than bumetanide) and robust anticonvulsant effects in preclinical rat models of drug-resistant epilepsy, although it does not clearly inhibit NKCC1 [150]. Conversely, KCC2 enhancers (e.g., the CLP257/CLP290 class) aim to increase Cl^−^ extrusion by stabilizing surface KCC2 and favoring phosphorylation states that promote transport activity. Robust in vivo efficacy has been reported in neuropathic pain, HD and AD models [9,151,152], but selectivity and target engagement require careful validation. CLP290 orally administered in mice (preclinical drug phase) during convulsant stimulations showed a better CNS penetration respect to bumetanide and a sustained restoration of the GABA inhibition, suppressing the epileptogenic process [153]. Modulators of Cl^−^ channels (e.g., benzodiazepines acting on GABA_A_ receptors) remain the cornerstone symptomatic treatment for epilepsy [17]. Emerging approaches include gene therapy to normalize NKCC1/KCC2 levels and strategies that modulate upstream regulators (e.g., BDNF, IGFs, WNK–SPAK/OSR1 signaling, Neurturin, inflammation) that set transporter activity [71,154,155,156]. Ultimately, combining transporter/channel-directed interventions with disease- and cell-type–specific modifiers—and adopting quantitative Cl^−^ imaging to stratify patients and confirm target engagement—will likely be required for durable benefit.

## 8. Conclusions

The maintenance of precise Cl^−^ homeostasis within neurons is fundamental for proper brain function, governing the polarity and strength of inhibition, neuronal excitability, and synaptic plasticity. Convergent evidence now implicates disturbed Cl^−^ regulation in multiple neurological disorders—including epilepsy, AD, ASD, HD, and neuropathic pain—via dysfunction or misregulation of key Cl^−^ channels and transporters (e.g., GABA_A_/glycine receptors, NKCC1, KCC2). These alterations shift [Cl^−^]_i_ and E_Cl_, thereby reshaping circuit dynamics and behavior. Although substantial progress has been made in mapping these mechanisms, disease- and cell-type–specific patterns and their temporal evolution remain incompletely understood.

This growing understanding has spurred the development of potential therapeutic strategies aimed at restoring or modulating Cl^−^ homeostasis, including pharmacological interventions targeting Cl^−^ transporters and channels. While initial clinical studies with compounds like bumetanide have shown promise in conditions like ASD and epilepsy, challenges remain in achieving effective access to the CNS and ensuring selectivity.

Future progress will likely depend on embracing precision medicine approaches that account for genetic variability and disease heterogeneity. The development of brain-penetrant, selective modulators, coupled with standardized protocols for measuring Cl^−^ homeostasis in vivo, will be crucial for translating preclinical findings into effective treatments [157]. Furthermore, emerging technologies like single-cell transcriptomics and spatial transcriptomics hold the potential to reveal cell-type-specific patterns of Cl^−^ dysregulation, identifying novel therapeutic targets and biomarkers. Specifically, future priorities include: rigorous, standardized in vivo quantification of [Cl^−^]_i_ and E_Cl_ across brain regions and cell types; development of brain-penetrant, selective NKCC1 inhibitors and truly selective KCC2 modulators, paired with CNS target-engagement biomarkers; and modulation of upstream regulators (e.g., WNK–SPAK/OSR1, BDNF/TrkB) with attention to context, sex, and cell type.

While significant progress has been made, further research is crucial to fully elucidate the complex role of Cl^−^ dysregulation in neuronal disorders and to translate these findings into effective clinical treatments that can ameliorate symptoms and improve the lives of affected individuals.

## Figures and Tables

**Figure 1 life-15-01461-f001:**
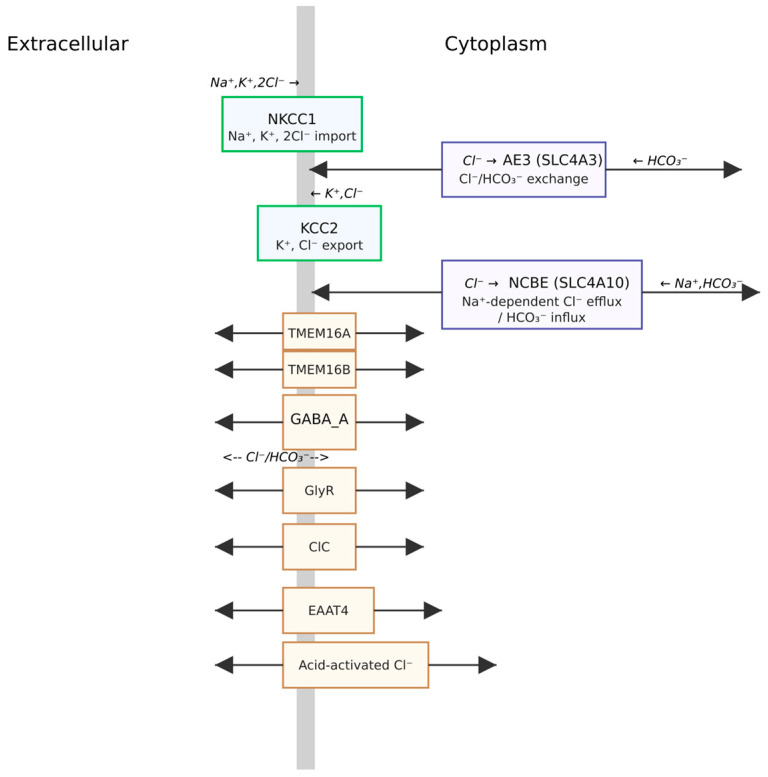
Integrated overview of neuronal Cl^−^ homeostasis. Passive Cl^−^ flux through GABA_A_ and glycine receptors, together with ClC channels and Ca^2+^-activated Cl^−^ channels (TMEM16A/B), interacts with active transport by NKCC1 (Na^+^, K^+^, 2Cl^−^ uptake) and KCC2 (K^+^, Cl^−^ extrusion) to set E_Cl_ and the polarity/strength of inhibition. AE3 (SLC4A3) and NCBE (SLC4A10) couple Cl^−^/HCO_3_^−^ exchange with intracellular pH regulation; EAAT4 and acid-activated Cl^−^ channels provide additional anion conductances. The net direction of ligand-gated Cl^−^ flow is determined by E_Cl_ (with GABA_A_ also conducting HCO_3_^−^).

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
