# Peer review of "Chloride Homeostasis in Neuronal Disorders: Bridging Measurement to Therapy"

_life, 2025, doi:10.3390/life15091461_

Round 1
Reviewer 1 Report
Comments and Suggestions for Authors
The review article by Arosio and Musio provides a broad and ambitious overview of neuronal chloride homeostasis and how this is involved in various neurological pathologies. Although the topic is relevant to basic and translational neuroscience, minor and major revisions must be addressed before publication.
Minor
Line 85. Abbreviations as HD, AD, and ASD are first shown in Table 1 without further definition for the general audience. Please, all acronyms or abbreviations must be defined at first use.
Line 175. Intracellular chloride concentration is first mentioned without the abbreviation ([Cl-]i) here, but on lines 190 and 250, it is shown with the acronym.
Line 128. The word lipofuscinosis has been written twice.
Line 151. The Na⁺-K⁺-Cl⁻ cotransporter 1 (NKCC1) lacks the 2 just before the Cl⁻ ion, meaning that NKCC1 is electroneutral.
Lines 85, 340, 341. If possible, add the appropriate bibliography or references to each line of the Tables.
Line 341. In the case of clinical trials or preclinical studies, identifiers (when available) or publications would be helpful.
Lines 342, Section 5.1. Consider including Delpire's group references on how KCC2 and NKCC1 deficiencies lead to epilepsy in rodents.
Line 435. To maintain consistency throughout the manuscript, the word co-transporters should be changed to cotransporters, without the hyphen.
Line 488. It refers to preclinical studies showing that CLP290 could be a treatment to enhance KCC2 activity and restore the excitation-inhibition balance and so, neuronal function. However, the reference is a review that only superficially addresses the topic. Why not cite the original research work instead?
Major Concerns
Some sections lack a cohesive description. For example, section 2 discusses many chloride channels without integrating them or explaining how they might interact with KCC2 or NKCC1. It also lacks a discussion about functional compensation in the absence of other channels or cotransporters. Thus, authors need to clarify their interplay and how the chloride dynamics are coordinated.
There is limited explanation of how NKCC1/KCC2 are regulated, mainly by knowing that the WNK-SPAK/OSR1 cascade is central to this family of proteins. There is also a lack of discussion regarding the presence of exchangers like AE3 and NCBE, and how they interact with cotransporters or respond to neuronal activity.
Throughout the manuscript, there is no mention about sex differences regarding chloride regulation. This is relevant in pain and neurological development investigations.
Authors suggested that chloride may act as a signaling molecule; however, the idea is not well developed. Thus, it would be useful to add examples or discuss potential drug implications.
Although there is a comprehensive description of individual available methodologies to estimate chloride dynamics, there is not enough discussion or comparison across methodologies. ClopHensor relies on calibration curves and controls, but this is not discussed for MQAE or SuperClomeleon. By using these techniques, it is possible to distinguish chloride dynamics among cell types, or is there a method more effective to estimate chloride dysregulation in animal models with AD, HD, ASD, or epilepsy?
The discussion about thallium and sodium flux assays is limited.
There is a lack of mechanistic details on how CLP290 acts over KCC2 and how bumetanide, an inhibitor of NKCC1, can reach the CNS, their interactions, and limitations. It would be helpful to include or summarize clinical trials to better understand and discuss existing controversies.
There is no mention of any compensatory roles of KCC3 or even other cation-chloride cotransporters in vivo; CCCs are expressed throughout the CNS.
Some sections are based on only one publication, and the information is contrary to that of other authors. For example, Section 5.2 is based on Keramidis et al. 2023, stating that Aβ42 treatment caused a downregulation of KCC2, and just a year before, Lam et al in Molecules reported that injection of Aβ42 in the mouse brain does not affect KCC2 expression, but NKCC1 expression increases after 30 days. Why avoid the discussion; what is the authors opinion?
A figure illustrating the role of NKCC1/KCC2 cotransporters in healthy and diseased neurons would be helpful. Another figure showing all the involved chloride channels is also relevant.
Comments on the Quality of English LanguageIn general, the manuscript is comprehensible but there are several cases of awkward phrasing, and grammatical inconsistencies. Thus, it could be very much improved for clarity, conciseness, and flow with the help of a native English speaker or professional scientific editor.
Author Response
See uploaded PDF.

Reviewer 2 Report
Comments and Suggestions for Authors
Please find attached the review comments for the article titled "Chloride Homeostasis in Neuronal Disorders: Bridging Measurement to Therapy."

Further details can be found in the attached file.
Author Response
See uploaded PDF.

Reviewer 3 Report
Comments and Suggestions for Authors
General comments:
This manuscript summarizes the findings of preclinical and clinical studies describing the role of chloride homeostasis for normal brain function, its disruption in diverse neurological disorders, and provides a link to restoring chloride homeostasis as a potential treatment. This review is comprehensive, explaining first the mechanisms maintaining neuronal chloride homeostasis, e.g., chloride channels and co-transporters activities. Second, different methods and techniques –from traditional to novel ones– to measure chloride concentration and dynamics. Third, provides an overview of how diverse mechanisms contributing to chloride dysregulation can be associated with several neurological disorders. Lastly, it briefly discusses potential therapeutic strategies investigated in the literature.
There is plenty of literature that describes the role of chloride cotransporters for normal neuronal function as well as the role of chloride homeostasis for neurological disorders, including epilepsy, autism spectrum disorder, Alzheimer’s disease, and schizophrenia, which constitutes an overall weakness of this review. A strength, however, is that the review includes an up-to-date summary of the methods and advanced techniques to measure neuronal chloride concentration, including fluorescence imaging techniques, and touches upon the more recent therapeutic strategies for neurological conditions targeting chloride homeostasis. Accordingly, the review provides an integrated overview.
Specific major comments
- Table 1. GABAA receptors' main function in neurons: unclear if it refers to E/I ratios or if it can be both excitatory and/or inhibitory. If so, then please specify the developmental stage associated with each function. Please be specific regarding the observed impairments. Replace “altered” with the corresponding disturbance (e.g., increased or decreased, reduced or overexpressed, etc.). For the Glycine receptor, enumerate which are the related mutations in the “specific impairment column” (e.g., GLRA1 or GLRB genes). Same for the ClC channels, briefly enumerate some related mutations with epilepsy, ataxia, etc.
- Page 5, line 198: “In mature neurons, the KCC2 cotransporter maintains a low [Cl⁻]ᵢ, causing GABAA and glycine receptors activation to drive Cl⁻ efflux and membrane hyperpolarization”.
In mature neurons, GABAA activation and glycine receptors drives Cl⁻ influx and thus, hyperpolarization, not efflux . Please change.
- Page 11. Huntington’s Disease, line 418. The reference #114 (Dargaei Z, Bang JY, et al., GABAergic inhibition rescues memory deficits in a Huntington's disease mouse model. Proc Natl Acad Sci U S A. 2018 Feb 13;115(7):E1618-E1626. doi: 10.1073/pnas.1716871115). States that restoring chlroide homeostasis can restore the cognitive deficits (learning and memory) in HD. Not the motor deficits.
On the other hand, reference #13 (Serranilla, M.; Pressey, J.C.; Woodin, M.A. Restoring Compromised Cl− in D2 Neurons of a Huntington’s Disease Mouse Model 559 Rescues Motor Disability. J. Neurosci. 2024, 44, doi:10.1523/JNEUROSCI.0215-24.2024) does suggest that treatment with bumetanide can delay the motor deficits in R6/2 mice, and that overexpressing KCC in D2 MSNs of R6/2 mice also improved motor function.
Please revise the reference or change accordingly.
Specific minor comments
- Abstract, line 12-17: for each disorder, specify which findings are in animal models and which have been investigated in humans.
- Introduction, page 2, line 74: “holds significant promise for improving the lives of individuals” can be seen as an overstatement, particularly given the variability in treatment response with current therapies aiming to restore chloride homeostasis. Please rephrase moderating this statement.
- Page 7, line 289: Specify what “CFP and YFP” stand for. These fluorescent abbreviations have not been described before.
- Page 8, line 315: “exceptional brightness…and superb photostability”. Instead of using the qualifying adjectives “exceptional” and “superb”, please describe more objectively how the brightness and photostability form the encoded voltage indicators (GEVIs), like Vega, are superior to others.
- Page 8, line 323: for the thalium flux essays, consider also the reference: Zhang S, Meor Azlan NF, Josiah SS, et al., The role of SLC12A family of cation-chloride cotransporters and drug discovery methodologies. J Pharm Anal. 2023 Dec;13(12):1471-1495. doi: 10.1016/j.jpha.2023.09.002. Epub 2023 Sep 9. PMID: 38223443; PMCID: PMC10785268. Since it describes in detail their ability to measure the SLC12
- Table 2. Please add the corresponding supporting citations at the end of each row (helps the reader and avoids having to look for them in the main text). Avoid ambiguous terms such as dysfunction, altered, or imbalance. Please describe the direction of the dysfunction for each disorder.
Example:
Epilepsy row, change KCC2 dysfunction for “reduce or loss”. The term “weakend/reversed GABAergic inhibition” might be confusing. Please rephrase to e.g., “reduced GABAergic inhibition and hyperexcitability”.
Autism row: describe the imbalances, i.e., increased expression on NKCC1 and reduced KCC2. And for the third column (impact in neuronal excitability) describe if it is reduced inhibition, decreased excitation, or both? And specify what does “altered” GABAergic signaling mean. The same goes for Huntington’s disease.
10. Table 3. Include the supporting references in the fourth or fifth column (current status or key findings). First row (NKCC1 inhibitors) sentence in last column is not clear: Please rephrase and add “variability in treatment response”. For the Gene therapy row. Please specify examples for “Various” in the third column (disorders of interest). Add a legend explaining the abbreviations used for ASD and AD, or spell them out.
11. Section 5.2 (Alzheimer's Disease (AD): A Gradual Fading of Memory). Line 367. For the sake of completeness, add how APP can modulate KCC2 Expression and Function in Hippocampal GABAergic Inhibition (references: Chen, M.; Wang, J.; et al. APP Modulates KCC2. Expression and Function in Hippocampal GABAergic Inhibition. eLife 2017, 6, e20142. and Doshina, A.; Gourgue, F.; et al. Cortical Cells Reveal APP as a New Player in the Regulation of GABAergic Neurotransmission. Sci. Rep. 2017, 7, 370).
12. Page 11. Line 379. Specify (give examples) which pharmacological agents have shown to reverse cognitive decline.
13. Page 11. Autism Spectrum Disorder (ASD) Line 391. Add reference to: Tang, X., Jaenisch, R. & Sur, M. The role of GABAergic signalling in neurodevelopmental disorders. Nat Rev Neurosci 22, 290–307 (2021). https://doi.org/10.1038/s41583-021-00443-x A recent review that describes in detail the disturbances in expression of NKCC1 and KCC2 and how it affects GABA signaling in neurodevelopmental disorders.
14. Page 12, lines 458-476. (6. Emerging Research on Chloride Dysregulation in Symptom Pathogenesis). This section seems repetitive with what has already been mentioned for each disorder in section 5. Consider adding/merging this section with the one above.
Author Response
See Uploaded pdf.

Round 2
Reviewer 1 Report
Comments and Suggestions for Authors
The corrected manuscript provides a more precise, clear, and comprehensive overview of neuronal chloride homeostasis and its role in various neurological disorders.
Minor.
Line 56. Add a space between AD's and models
Line 154. A space is missing between thereby and influence
Line 166. Remove the word AS
Line 239. The p is missing in the text
Line 443. A space is missing between with and impaired
Line 564. The title must be similar to the rest: only use capital letters on the first word
Line 596. Remove the i before neurological disorders
Line 615. ECl must be changed to ECl
Remove the bold in Table 3 to look like the rest
Please review for misspellings and other minor details.
Author Response
See attached PDF.

Reviewer 2 Report
Comments and Suggestions for Authors
I believe the authors have significantly improved the overall quality, clarity, and flow of the manuscript compared to the previous version. The revised text is more coherent and easier to follow. However, a few minor issues remain, which should be addressed to enhance the manuscript’s consistency and readability further:
- Abbreviations: Despite previous comments regarding inconsistent use of abbreviations, several errors remain uncorrected. Terms such as [Cl⁻]ᵢ, Cl⁻, ECl⁻, AD, and HD are still not properly defined upon first use, and their usage is inconsistent throughout the manuscript. I strongly recommend a thorough revision to ensure that all abbreviations are introduced clearly when first mentioned and used consistently thereafter.
- In line 154, please add a space between “thereby” and “influence”.
- In section 2.3, it would be appropriate to expand this section by including a brief expansion, in just a few lines, about other trophic factors beyond BDNF. The authors may use reference 156, already cited in the text, to provide additional detail. This would help contextualize their potential roles and provide a more comprehensive overview of the signaling environment under discussion.
Author Response
See attached PDF.

Reviewer 3 Report
Comments and Suggestions for Authors
Major Comment #1:
I thank the authors for carefully addressing all my comments, however, this revised version has an important misalignment in the references that needs to be thoroughly revised; the way it is currently is incorrect starting from page 13 and onward (reference #126 until ref #162).
In-text references seem to be shifted, starting in section 5.3 Autism Spectrum Disorder (page 13, line 457) with Reference #126 Doshina et. al., 2017 until page 15 reference #141.
Example: on page 13, line 457, Ref # 126 refers to Alzheimer disease, not to ASD (it seems that right reference should be #128 Tang X et al.,). Similarly, references in line 482 [#126 and 127] should be refs #129,130 (for autism). In section 5.5 Other Neurological Disorders. The cited references for Down syndrome are # 128-130 (which refer to Autism and oxytocin for autism), when they should be refs # 132,133 (Deidda et al. 2015 which refers to autism).
Also, from reference #149 in page 16, line 582 until page 17 reference #154 the references are also incorrect (the also seemed to be shifted). References in table 3. #155-159, are also misaligned with the therapeutic strategy (e.g. in benzodiazepines row, ref #158 and for oxytocin #159).
Please revise all the references accordingly.
Minor comments:
Comment #1
Table 1. There is a typo on the first column title: It now says “Table. Cont.” instead of “Transporter/channel name”. Please revise.
Comment #2
Section 7. Page 16, line 574: Please specify that bumepamine has been tested in preclinical models (rat models of epilepsy).
Comment #3
There is a typo in Section 7. Page 16, line 578. It currently says ‘Robust in vivo efficacy has been reported for neuropatic pain and AD”. AD stands for Alzheimer’s disease. It should be ASD (autism spectrum disorder) according to reference #147.
Comment #4
Section 7. page 16, line 579. “ CLP290 orally administered in mice (preclinical drug phase) during convulsant stimulations showed a better CNS penetration respect to bumetanide and a sustained restoration of the GABA inhibition, suppressing the epileptogenic process [149].”
Reference # 149 is for Bumepamine, and it does not mention CLP290. Please revise the reference (seems that the correct one is #152, Cai J 2024.).
Author Response
See attached PDF.
